# Performance of a new co-axial ion-molecule reaction region for low-pressure chemical ionization mass spectrometry with reduced instrument wall interactions

Brett B. Palm[1], Xiaoxi Liu[2], Jose L. Jimenez[2], and Joel A. Thornton[1]

[1]Department of Atmospheric Sciences, University of Washington, Seattle, WA, USA
[2]Department of Chemistry and Cooperative Institute for Research in Environmental Sciences, University of Colorado, Boulder, CO, USA

Correspondence: Joel A. Thornton (joelt@uw.edu)

**Abstract**

Chemical ionization mass spectrometry (CIMS) techniques have become prominent methods for sampling trace gases of relatively low volatility. Such gases are often referred to as being "sticky," i.e. having measurement artifacts due to interactions between analyte molecules and instrument walls, given their tendency to interact with wall surfaces via absorption or adsorption processes. These surface interactions can impact the precision, accuracy, and detection limits of the measurements. We introduce a low-pressure ion-molecule reaction (IMR) region primarily built for performing iodide-adduct ionization, though other adduct ionization schemes could be employed. The design goals were to improve upon previous low-pressure IMR versions by reducing impacts of wall interactions at low pressure while maintaining sufficient ion-molecule reaction times. Chamber measurements demonstrate that the IMR delay times (i.e., magnitude of wall interactions) for a range of organic molecules spanning five orders of magnitude in volatility are 3 to 10 times lower in the new IMR compared to previous versions. Despite these improvements, wall interactions are still present and need to be understood. To that end, we also introduce a conceptual framework for considering instrument wall interactions and a measurement protocol to accurately capture the time-dependence of analyte concentrations. This protocol uses short-duration, high-frequency measurements of the total background (i.e., fast zeros) during ambient measurements as well as during calibration factor determinations. This framework and associated terminology applies to any instrument and ionization technique that samples compounds susceptible to wall interactions.

## 1    Introduction

Trace gases in the atmosphere are drivers of the chemistry that determines air quality and climate effects (Seinfeld and Pandis, 2006), as well as oxidant budgets and oxidation pathways (e.g., Crutzen, 1979; Di Carlo et al., 2004) and secondary organic aerosol (SOA) formation (Shrivastava et al., 2017). Trace organic compounds are particularly complex, spanning more than 15 orders of magnitude in volatility (Donahue et al., 2012; Hunter et al., 2017; Isaacman-VanWertz et al., 2018). Large gaps remain in our knowledge of the chemistry and impacts of trace organic gases, in particular the lower volatility compounds (Goldstein and Galbally, 2007). The ability to measure and quantify such lower volatility gases is an evolving analytical measurement challenge, but remains a limiting factor in our ability to test important theories governing, e.g., organic gas-particle partitioning, oxidation mechanisms, SOA formation, vertical distributions, and dry deposition.

Many of the recent advances in knowledge of atmospheric trace gases, particularly the lower volatility compounds in the gas phase, have been due to the application and development of advanced instrumentation (Mohr et al., 2013; Ehn et al., 2014; Isaacman et al., 2014; Krechmer et al., 2016a; Peng et al., 2016; Yuan et al., 2017). One of these major advances has been the development of field-deployable mass spectrometers, combined with the development of specialized inlets allowing the application of various chemical ionization methods to atmospheric compounds (Ehn et al., 2014; Lee et al., 2014; Krechmer et al., 2016a). In chemical ionization, analyte molecules are imparted an electrical charge either by charge transfer from or clustering with a reagent ion, processes which are relatively low energy and typically induce little fragmentation of the analyte molecules. A variety of reagent ions with the ability to ionize different subsets of analyte molecules have been used, including $H_3O^+$, acetate, iodide ($I^-$), nitrate ($NO_3^-$), ammonium ($NH_4^+$), and others (e.g., Jokinen et al., 2012; Yatavelli et al., 2014; Zaytsev et al., 2019). Iodide-adduct ionization in particular has been used for both gas and particle composition measurements, and is sensitive to a wide range of inorganic and organic molecules (e.g., Huey et al., 1995; Le Breton et al., 2012; Lopez-Hilfiker et al., 2013; Mohr et al., 2013; Lee et al., 2014; Veres et al., 2015; Gaston et al., 2016; Lee et al., 2016).

One impediment to the measurement of lower volatility gases is the influence of inlet tubing and other experimental apparatus surfaces. Several recent experiments have probed the effects of Teflon chamber walls on experimental processes (Matsunaga and Ziemann, 2010; Krechmer et al., 2016b; Krechmer et al., 2017;

Huang et al., 2018). A variety of organic and inorganic gases have been shown to reversibly absorb into Teflon
(and other polymer) tubing or reversibly adsorb onto the surface of a variety of solid materials including
stainless steel (Pagonis et al., 2017; Deming et al., 2019; Liu et al., 2019). Current CIMS instrumentation typically
requires the use of such materials in the design of the inlet tubing as well as the ion-molecule reaction (IMR)
region where chemical ionization occurs, which allows for wall interactions to occur. The rates of flux of analyte
molecules to and from these wall surfaces can depend on complex factors of water vapor concentration, co-
analyte concentrations, etc. (Pagonis et al., 2017; Deming et al., 2019; Liu et al., 2019), leading to difficult
interpretations of data that is often not consistent across different studies.

Past CIMS IMR versions have employed different designs, typically constructed from varying fractions of

stainless steel and several types of Teflon (Eisele and Tanner, 1993; Jokinen et al., 2012; Lee et al., 2014; Zhao et
al., 2017; Lee et al., 2018). Some IMR designs, such as the $NO_3^-$ CIMS, can operate at ambient pressure with an
IMR design that essentially eliminates wall interactions (e.g., Krechmer et al., 2015). However, the $NO_3^-$ reagent
ion is sensitive only to a narrow subset of highly-oxidized molecules with which it has clustering strengths
greater than its cluster with $HNO_3$. The $I^-$ CIMS technique is sensitive to a much broader range of analyte
molecules, making it a powerful technique for studying atmospheric chemistry. But, $I^-$ can also cluster with one
or more water molecules, causing the sensitivity of $I^-$ toward other analyte molecules to be dependent on water
vapor concentrations in the IMR. To reduce this water vapor dependence, the IMR is typically operated at low
pressure (~2-200 Torr) to reduce the partial pressure of water vapor. For aircraft $I^-$ CIMS measurements, a low-
pressure IMR has also been desired in order to allow pressure control systems to maintain constant pressure in
the ionization region with changing pressure/altitude, thus maintaining constant sensitivity to clustering
(Neuman et al., 2002; Crounse et al., 2006; Le Breton et al., 2012; Lee et al., 2018). In order to operate at low
pressure, the $I^-$ CIMS must sample through an orifice, necessitating wall interactions in the IMR. Accounting for
the flux of analyte from the IMR walls is a challenge of particular importance to aircraft measurements, where
ambient concentrations can change rapidly on the edges of spatially narrow plumes from point or regional
sources such as power plants, biomass burning, or urban areas. Background measurement and subtraction from
the total observed signal is typical (Neuman et al., 2002; e.g., Crounse et al., 2006; Veres et al., 2008; Lee et al.,
2018), however a uniform standard method for background subtraction does not exist, and methods applied by
different research groups vary widely.

In this work, we present a new design of a co-axial, low pressure IMR to minimize wall interactions,

incorporating knowledge acquired in the operation and analysis of past IMR designs. A detailed consideration of
the process of sampling through an instrument inlet is presented, explaining how the measured signal is
influenced by wall interactions. We suggest practices for accounting for wall interactions, both in experimental
measurements and when performing calibration measurements that will be later applied to experiments.
Finally, this new IMR design was characterized by measuring the magnitude of wall interactions of several
organic compounds spanning a wide range of volatility. Both the new IMR design considerations and the
broader discussion of wall interactions will be applicable to a broader community of analytical atmospheric
chemistry.
**2    Co-axial low-pressure IMR design**
**2.1    $I^-$ CIMS method**

Iodide-adduct chemical ionization has been described in detail in previous studies (Huey et al., 1995;

Lee et al., 2014; Lee et al., 2018). Briefly, $I^-$ anions are produced by passing methyl iodide ($CH_3I$) in nitrogen
through alpha particles from a polonium-210 radioactive sealed source. The anions form adducts by colliding
with neutral analytes inside an IMR, and the clusters are subsequently sampled by a high-resolution time-of-
flight mass spectrometer (HR-ToF-MS; Tofwerk AG, Thun, Switzerland). This spectrometer provides a nominal
mass resolving power ($m/\Delta m$) of approximately 5000 with pptv level detection limits for most compounds.
**2.2    IMR description**

Many different IMR designs have been employed in past CIMS measurements, each with advantages

and disadvantages. The primary function of any IMR region in a CIMS is to facilitate the process of imparting an
electrical charge onto analyte molecules in the sample air, whereupon they can be manipulated and analyzed
inside the mass spectrometer. Depending on which reagent ion is chosen and which analyte molecules are
targeted, the IMR will have different design requirements. Recent interest in identifying and quantifying a broad
range of reactive and/or low volatility compounds presents substantial challenges for CIMS instruments with
low-pressure ionization regions, including but certainly not limited to the $I^-$ CIMS used in this work. The effects
of IMR wall interactions can be a substantial impediment to making accurate and easily interpretable
measurements of compounds that react on or reversibly partition to reactor walls.

Herein we describe the design of a new co-axial IMR, illustrated in the schematic in Fig. 1. This design aimed to improve upon that most recently employed by the Thornton research group during the WINTER 2015 research flights, which has been described in detail in Lee et al. (2018). That version was itself a design built to improve upon the characteristics of previous versions of the IMR including the model available commercially from Aerodyne Research Inc. with the mass spectrometer (Kercher et al., 2009; Bertram et al., 2011; Lee et al., 2014). In the commercially available low-pressure IMR, the analyte flow and ion flow are mixed via turbulence inside a region constructed out of stainless steel. In addition to the increased wall interactions that result from turbulence, stainless steel has been shown to suffer from enhanced wall effects for many compounds (Deming et al., 2019; Liu et al., 2019). The WINTER IMR made improvements by decreasing the wall surface area and residence time of the turbulent region, and also by constructing two of the three walls of the cylindrical IMR region out of machined PTFE Teflon (Lee et al., 2018). However, the third wall remained stainless steel, and turbulence remained an issue.

The main goals of our improved IMR design were to reduce wall effects while maintaining sufficient residence time for clustering (i.e., maintain sufficient sensitivity). The initial strategies were to remove as many wall surfaces as possible, and have any necessary wall surfaces be constructed from materials such as perfluoroalkoxy (PFA) Teflon which have been shown to have the weakest interactions with many analytes (Pagonis et al., 2017; Deming et al., 2019; Liu et al., 2019). To minimize wall effects further, we aimed to inject the sample flow into a co-axial sheath of ion flow, creating a larger distance between analyte and surfaces. This design feature was similar to what has been used in some previous IMR designs, in particular for the $NO_3^-$ reagent ion (Eisele and Tanner, 1993; Jokinen et al., 2012; Massoli et al., 2018). Furthermore, we aimed to pump the flow out of the IMR in a similar manner to how it was injected, pumping a sheath flow radially outside of a sample flow. Any analyte that desorbed from a wall surface would be more likely pumped out in the sheath flow and not sampled into the MS.

The final design requirement was that the IMR was capable of operating at a constant IMR pressure on an aircraft platform, where ambient pressure can span the range from ~200-760 Torr. Ion-molecule reaction rates scale with total analyte number density, and ion-molecule cluster stability will depend on total pressure as well as $H_2O$ partial pressure (Lee et al., 2014; Iyer et al., 2016; Lopez-Hilfiker et al., 2016). Thus, maintaining constant pressure (and temperature) can minimize changes in instrument response with large changes in

altitude. This feature was added to the WINTER version of the IMR by incorporating a variable orifice on the upstream side of the IMR (Lee et al., 2018), and it is also included in this new co-axial IMR. As long as the pressure downstream of the orifice remains roughly less than half of the pressure of ambient air upstream, critical flow is achieved in the orifice (i.e., the speed of the air through the orifice is approximately the speed of sound). The mass flow through the orifice is then only a function of upstream pressure. As upstream pressure changes with altitude, the variable orifice can be opened or closed via computer control to maintain constant mass flow into the IMR. As the pumps maintain constant mass flow out of the IMR, the pressure inside the IMR remains constant at ~70 Torr downstream inside the IMR where I⁻ is introduced and ionization occurs.

The benefits of constant reduced pressure, e.g. stable instrument response and reduced effects of water vapor on ionization efficiency, come with enhanced wall interactions which can contribute potentially large and often poorly understood artifacts to the measurement. The pressure drop between ambient pressure and ~70 Torr leads to a high velocity jet expansion, which induces turbulent mixing. The jet-induced turbulence ensured mixing of reagent ions and sample flows in previous IMR designs, but also enhanced contact of the sample flow with IMR surfaces. Moreover, the low pressure leads to an order of magnitude larger diffusivity compared to ambient pressure, such that even in the absence of jet induced turbulence, gases in the sample flow will randomly reach the walls of the IMR more efficiently than at typical ambient pressures. Consistent with these ideas, it has been previously shown that the low pressure IMR is the main source of instrument memory and reactive trace gas losses, not the ~0.5 m long sampling inlet at ambient pressure with fast (~10-20 slpm) flow rates typically used (Lee et al., 2018).

Given the above considerations, the first design challenge was to slow the sample flow rate down by expanding the flow cross section while limiting turbulent mixing of analyte molecules to wall surfaces. In order to expand the flow without causing turbulence, an expansion cone/diffuser with an angle of less than approximately 5–7 degrees could be used. Fluid dynamics simulations have shown that this method can prevent flow separation that leads to turbulence in expansions, though possibly not for the Reynold's numbers of less than 2000 in this IMR (Sparrow et al., 2009, and references therein). This cone angle would require a length of more than 13 cm. Diffusion calculations suggest that one third of the analyte would contact the diffuser wall surface under laminar conditions, which still requires getting the flow laminar after the orifice. Given these considerations, as well as time constraints prior to a field campaign, we opted not to test a conical diffuser at

this time. Instead, the jet of air exiting the orifice was allowed to expand immediately into a fluorinated

ethylene propylene (FEP) Teflon-lined cylinder with 1.2 cm diameter and 1 cm length, after which it passed

through a parallel cluster of 3.175 mm OD, 1.5875 mm ID (0.125 inch OD, 0.0625 inch ID) PFA Teflon tubes with

a length of 1.5 cm. Turbulence was limited to the 1.2 cm diameter cylinder, and then the subsequent tubing

cluster acted to develop laminar flow. As a rough approximation, turbulent flow can be converted to laminar

flow by passing through a tube with an entrance length that is 10 times its diameter (Çengel and Cimbala,

2014). This concept guided our design specifications. When the sample air exits the laminizer element, the flow

has slowed down and become much less turbulent, mitigating the effects of walls downstream of that point.

Since having an orifice upstream of the IMR effectively necessitates having some region of turbulence in contact

with walls, this design strategy was aimed at limiting the residence time and amount of wall surface area in the

region of the IMR that encountered turbulent sample gas. Future low-pressure IMR designs could aim to further

minimize wall effects in this region directly downstream of the variable orifice.

While the sample gas enters the IMR through the orifice, the I$^-$ anions are concurrently injected into a

region of the IMR concentric with and outside of the sample flow laminizing element. The anions are produced

by flowing dry $N_2$ over a permeation tube containing methyl iodide and then through a Po-210 radioactive

sealed source, producing I$^-$. The ion flow experiences some turbulence when injected into the IMR, and then

passes through a parallel cluster of 6.35 mm OD, 3.175 mm ID (0.25 inch OD, 0.125 inch ID) PFA Teflon tubes

with a length of 1.27 cm (0.5 inch) that act as a laminizer element. The flow coming out of both the sample flow

laminizer and ion flow laminizer exit in the same plane and can be arranged to have approximately the same

velocity in the axial direction. In this work, this was achieved by maintaining a constant 2 slpm sample flow and

3 slpm ionizer flow. As part of the process of designing the IMR with laminizers, fluid modeling simulations were

performed to visualize the effects of turbulent vs. laminar flows. Two example cases are depicted in Fig. S1.

The exit of the laminizers marks the start of the drift region in the IMR where interactions of analyte

with I$^-$ anions occur. Within the 3.49 cm (1.375 inch) ID, 3.81 cm (1.5 inch) long drift region, the I$^-$ anions and

analyte flows mix together via diffusion, possibly aided by some residual turbulence. The design also includes

some exposed stainless steel surfaces on the drift region wall and at the exit of the sample flow laminizer and

entrance of sample pump flow tube, as far from the main sample flow as possible to limit wall interactions.

These surfaces can be used to apply an electric field inside the IMR to attempt to enhance the mixing of ions

into the sample flow. However, only modest total detected ion enhancements were measured when applying

such electric fields. We hypothesize that the relatively high diffusivity at 70 Torr, as well as any residual

turbulence, were dominating the flow mixing instead of the electric field forces in this particular design.

Because of the only modest gains and in the interest of simplicity, all exposed metal surfaces were grounded

together and electric fields were not employed during the measurements discussed herein.

Because the analyte molecules enter the drift region in the center, they would have to diffuse all the

way across the ion flow to reach a wall surface. In order to be sampled after encountering a wall, they would

also have to diffuse all the way back across the ion flow to the capillary into the mass spectrometer. To prevent

any molecules coming from the drift tube wall being sampled, half of the drift tube flow was pumped out along

the drift tube wall and away from the MS capillary. According to diffusion calculations, only 4% of the analyte

are predicted to encounter a wall in the drift region under laminar flow conditions, and a small fraction of those

molecules would diffuse back to the center to be sampled, essentially removing the effects of the drift region

walls. The other half of the drift region flow was pumped through an FEP Teflon-lined sample tube with ID of

2.18 cm (0.86 inch) and length of 5.08 cm (2.0 inch) and past the MS capillary, where it was sub-sampled into

the mass spectrometer.

Limiting the interaction between analyte and wall surfaces also limits the possibility of the analyte

undergoing chemical reactions on surfaces. To examine and quantify the improvements made in this design, we

start with a comprehensive discussion of the origin and meaning of wall effects. Although wall interactions are

not the only source of instrumental background signals, for semi-volatile and low volatility compounds they are

often the dominant source of residual non-ambient signal. The concept of background signal will be examined

using laboratory measurements, and further discussed in the context of ambient measurements and instrument

response calibrations. The improvements will be assessed by comparing laboratory measurements made with

this IMR to previous measurements from other IMRs and instruments.

**3    The effects of instrument wall surfaces**
**3.1    Measuring and subtracting instrument background signal**

In order to properly evaluate the new IMR design, we must first introduce a common framework that

can be used to describe how inlet tube and IMR wall interactions originate, what their effects are, and how they

can be understood. The CIMS experimental setup will be defined here as comprised of two parts: the sampling

tube (i.e., inlet) which transports the analyte from the sampling location (outside of aircraft, inside chamber, etc.) to the IMR; and the IMR, where ionization occurs prior to entering the MS. The IMR is defined as part of the instrument. The background signal is typically measured by flooding the sampling tube and/or IMR with clean air or ultra-high-purity nitrogen (UHP $N_2$). Subtracting the resulting "background signal" from the total signal measured while sampling ambient air is a common practice in atmospheric mass spectrometry. However, the exact definition and quantification procedure of the 'background' can vary across different experimental configurations and analysis goals. The processes that lead to the background signal can also be dynamic and controlled by multiple factors.

The background signal can originate from molecules coming from either the sampling tube or the IMR. In many cases, the sampling tube can be designed such that its background effects are small relative to the IMR effects, e.g., by pulling a large flow through the inlet and subsampling into the IMR, thus minimizing inlet residence time and also diluting the flux from the walls into a large flow volume. Sampling at ambient pressure in the sample tube also minimizes diffusivity to and from the walls. The IMR walls have been shown to be the dominant source of background signal in previous field measurement setups (Lee et al., 2018), so this discussion will focus mainly on IMR background signal. The details and concepts discussed here of background signal sources and how to quantify them are not specific to the I⁻ CIMS IMR, but can be adapted to other IMRs and ionization types as well as for sampling tubes. The concepts involved are illustrated in Fig. 2a and demonstrated using laboratory measurements in Fig. 2b, where a constant gas-phase concentration of nitric acid ($HNO_3$) was injected into a short polytetrafluoroethylene (PTFE) Teflon inlet tube (~20 cm length, 0.75" diameter, 20 slpm flow rate) and subsampled into the IMR in the sample flow for a specified amount of time. The effects of wall interactions in such an inlet are minor relative to the effects of wall interactions inside the IMR (as demonstrated in Fig. 2). The schematic in Fig. 2a and the following discussion applies mainly to analyte molecules that partition reversibly to the walls (or to thin films of water adsorbed on the walls, as is the case for $HNO_3$; Liu et al., 2019), and for wall surfaces that allow for absorption such as Teflon varieties. Adsorbing surfaces such as stainless steel, and irreversible loss of analytes such as many radical species, will be discussed as exceptions.

At the theoretical time t=$t_0$ in Fig. 2, consider an IMR that has never previously sampled a specific analyte molecule in the sample flow. Prior to $t_0$, there will be no signal at all from this analyte entering in the

sample flow, and the only signal corresponding to that analyte will be defined here as the persistent
background, due to electronic noise and other baseline signal sources such as the ion source or carrier flows. In
the specific case of $HNO_3$ in the Fig. 2b example, a substantial persistent background exists due to a source in
the ion flow from the Po-210 ionizer. Most analytes will not have such a persistent background. At $t=t_0$, the
analyte has entered the IMR and experienced one of the following two fates: 1) traveled directly from outside
of the IMR to the detector in the gas phase without interacting with a wall surface (which may include bouncing
off of a wall surface without interacting), or 2) absorbing in (or adsorbing on) a wall surface, where it remains
for some amount of time longer than the residence time of the IMR before desorbing and being sampled to the
detector. The fraction of analyte that follows each of these two paths will be a function of instrument design
(i.e., what fraction of sampled air collides with a wall surface through turbulence or diffusion) and as a function
of the uptake and partitioning coefficients of each analyte on each wall surface type. The uptake coefficients
themselves will be a function of the exact environmental conditions of the wall surfaces at the time of collision.
These environmental conditions can modify the wall surfaces and change how gases are taken up into/on
surfaces or change how they desorb from the surfaces.
The most influential surface modifier is often water. The analyte can behave differently depending on
whether it encounters a bare Teflon or stainless steel surface under completely dry conditions, a surface coated
in a monolayer of water under low RH conditions, or a surface coated with a thick layer of water that causes an
aqueous diffusion limitation to the analyte interacting with the actual surface. Liu et al. (2019) demonstrated
that some polar compounds partition to walls as a function of their Henry's Law constants during humidified
conditions. This IMR design has the ability to add water vapor directly downstream of the variable orifice as in
Lee et al. (2018). This maintains a relatively narrow range of water vapor concentrations in the IMR regardless
of the sample air humidity, keeping the environmental conditions (and uptake/partitioning coefficients) in the
IMR roughly constant. Surfaces can also be modified by other analyte molecules, which essentially act in
competition for surface sites. This behavior has been observed for materials such as stainless steel that are
dominated by adsorption to a limited number of surface sites (Deming et al., 2019). While absorbing materials
such as Teflon have been shown to be modified by water, they appear to be insensitive to the amount of other
analytes absorbed in the surface (Pagonis et al., 2017; Deming et al., 2019; Liu et al., 2019), at least at analyte
concentrations relevant to the atmosphere and typical laboratory chamber experiments.

As soon as there are analyte molecules ad/absorbed on surfaces, there will be a flux of that analyte from the surface back into the sample/ion flow. The flux from the surface will be a function of the amount of analyte on the surface, as well as the environmental conditions such as temperature, humidity, and history (see above). Therefore, in the moments just after $t=t_0$, e.g. $t=t_1$ in Fig. 2, there will be a flux of analyte from the walls. We define this flux as the source for the dynamic background signal, which is separate from the persistent background signal. Any analyte that is entering the IMR at this time will continue to split between reaching the detector directly or absorbing into the walls first, at the same fractional rates. These fractional rates will be constant as long as the environmental conditions remain constant, and the rates will not be a function of the flux of that analyte coming off of the wall. At $t=t_1$, the total flux into the IMR is greater than the total flux to the detector, and there is a net flux to the wall surfaces. As more analyte continues to enter the inlet and ad/absorb on the walls, the flux of analyte from the wall will continue to grow until a time comparable to $t=t_2$ in Fig. 2. Any analyte that partitions irreversibly to the walls or desorbs as a different compound due to surface reaction would appear to have no flux from the walls and no dynamic background signal. Only the fraction of such an analyte that did not interact with the walls would be detected, potentially at much lower sensitivities than expected from ionization efficiency considerations (Lopez-Hilfiker et al., 2016).

At times equivalent to $t=t_2$ in Fig. 2, the flux of reversibly-partitioning analyte from the wall has grown to be equal to the rate of ad/absorption of the analyte to the wall. The wall system is now in steady state. The amount of analyte arriving at the detector is now equal to the sum of the analyte that did not interact with walls and the analyte that entered the IMR at some earlier time, interacted with a wall, and then desorbed to reach the detector. Because the flux from the walls is equal to the flux to the walls, the total flux to the detector is equal to the total flux of analyte that is entering the IMR at that time. That is, the total signal is the same as it would be if the analyte were introduced into an IMR completely absent of wall interactions. This condition is only true when the incoming analyte concentration and environmental conditions have remained constant for long enough to establish wall steady state. As shown in Fig. 2b, the only signal that stays constant during a constant concentration injection with wall interactions is the background-subtracted signal. The background signal and thus also the total detected signal change over time and are both non-deterministically related to the analyte concentration entering the inlet. This concept is critical for the time-dependent quantification of analyte

in the sampled air, and is also important for the determination and interpretation of calibration factors, as
discussed later in Sect. 3.2.2.

The ratio of the background-subtracted signal to the background signal will remain constant after time

$t=t_2$, as long as the environmental conditions in the IMR remain constant. However, the ratio will not be the
same for all analytes. For analytes which are more volatile (or less soluble in water), interact with the wall
surfaces less strongly, and desorb more rapidly, the background signal may be negligible relative to the
background-subtracted signal (and the background-subtracted signal will be essentially equal to total signal).
For analytes which are less volatile (more soluble), interact strongly with surfaces, and desorb slowly, the
background signal may become a large majority of the total signal at the detector and the background-
subtracted signal may reach a detection limit. The IMR geometry and design will largely determine which
compounds qualify as 'more' and 'less' volatile on this relative scale. For instance, the $NO_3^-$ CIMS and the cross-
flow ion source (Zhao et al., 2010; Zhao et al., 2017) which operate with laminar flows at atmospheric pressure
in the IMR thereby minimizing turbulence and diffusion to walls, both employ geometries that prevent sampling
of any analyte that encountered a wall surface in the ionization region, leading to an IMR background signal flux
that is essentially negligible. Atmospheric pressure sampling, which as noted above is ultimately the source of
such benefits, may not be suitable for an aircraft platform as discussed above.

Continuing our description of the evolution of wall interactions, consider that the source of the analyte

into the IMR is completely removed immediately following $t=t_2$. For instance, this could represent the injection
of analyte-free air during a background measurement, or a scenario where the sampled air transitions rapidly
from high concentrations of the analyte in a plume to very low concentrations outside of a plume. There will be
a short transition period, corresponding to the residence time distribution of air in the IMR downstream of
where analyte-free air is injected (approximately 100 ms on average in the IMR described herein) plus any time
for switching flows outside the ionization region (potentially several seconds), when the analyte-laden air is
replaced with analyte-free air in the IMR. The flux of analyte to the detector without wall interactions and the
flux of analyte to the walls both drop to zero at this point, which is specified as $t=t_3$. There remains essentially
the same amount of analyte ad/absorbed on the walls at $t=t_3$ as at $t=t_2$ immediately prior, so the flux from the
wall to the detector continues to provide the same dynamic background signal.
After more time passes and $t=t_4$ has been reached, the amount of analyte on the walls has been
partially depleted since the wall system is now out of steady state. There is still a flux from the wall without a
complementary flux to the wall to replenish the analyte. The flux from the wall is also lower at this time than at
$t=t_3$ because the concentration of analyte on the wall is lower. At a subsequent time long after $t=t_4$, all of the
analyte would eventually desorb from the walls, and the dynamic background signal from the inlet walls would
reach zero, equivalent to a time $t<t_0$. As discussed further in Sect. 3.3, the amount of time required for the
dynamic background signal to decay to 10% of the original signal (i.e., to near zero) can range from less than 1 s
to tens of min or more, depending on the volatility of the analyte as well as environmental conditions and
surface types. For some analytes (including $HNO_3$ in the iodide anion source discussed here), there can be other
persistent sources of background signal coming from the tubing and carrier gas related to the ion source. The
persistent background is present at all times from $t<t_0$ to $t>t_4$, and can be quantified by injecting analyte-free air
for sufficient time to completely deplete the dynamic background signal of interest here. The persistent
background is also included in the signal measured during clean-air injections at $t=t_3$.
The main goal of measuring and subtracting the background signal in an instrument is to ascertain the
concentration of the analyte present in sampled air at the time of sampling with high temporal/spatial
resolution, removing the instrumental artifacts related to the background signal caused by wall interactions. As
illustrated in Fig. 2a, this task is often made complicated by the fact that the ratio of the background signal to
the background-subtracted signal can vary widely during measurements. The entire signal could be due to
background signal (as at $t=t_3$), due to gas phase signal ($t=t_0$), or some dynamic mix of the two ($t=t_1$ and $t=t_2$).
Even when all signal is coming from the background, the magnitude of the background can also change ($t=t_4$).
Given these fluctuating factors combined with a potentially rapidly changing sampling environment due
to a moving aircraft platform or rapidly shifting air masses with different source characteristics, the ideal way to
determine the true concentration of the analyte in sampled air is to measure the amount of signal coming from
the background sources at all points in time and subtract it from the total signal. But akin to Heisenberg's
uncertainty principle, one cannot precisely measure both the total signal and the background signal at the same
time. Instead, a practical method for determination of background signal is to measure the instantaneous flux
of analyte off the walls using high-frequency, short-duration injections of analyte-free gas (typically UHP $N_2$)
interspersed among the normal measurement of total signal, and then interpolate between these background

measurements. This method is has been referred to as performing 'fast zeros'. Upon injection of analyte-free gas, the measurement transitions from representing the total signal (equivalent to $t=t_1$, $t_2$, or $t_3$, depending on whether wall steady state has been achieved) to a measurement of just the sum of dynamic and persistent background signals (equivalent to $t=t_3$).

As seen in the inset of Fig. 2b, the decay of the analyte signal occurs in two parts (or more). The first part is the rapid exponential decay as the volume of the inlet is cleared out of any remaining gas-phase analyte, and stability of flows is achieved, etc. The next part, which applies when the analyte is of relatively lower volatility or higher Henry's Law constant into wall-adsorbed water, is the typically slower exponential decay that accompanies desorption of the analyte from the walls. There may be multiple decay constants with varying time scales (e.g., as illustrated in Krechmer et al., 2018) if there are multiple types of wall surfaces (e.g., both Teflon and stainless steel in the same IMR) or voids with different residence times. To know the background value at the time when the background measurement was initiated ($t=t_2$), one needs to know the magnitude at which the slower exponential decay begins, i.e. the signal value at $t=t_3$ shown in the inset of Fig. 2b. The background determined at successive times of $t=t_3$ are then interpolated to estimate the background at all points in time. Such periodic background determinations would also inherently account for any changes in the environmental conditions that would change the analyte uptake coefficient and thus the ratio of the flux to the walls vs the flux to the detector without wall interaction, such as an aircraft platform flying through varying ambient $H_2O$ concentrations. In other words, as long as the background signal can be determined at a given time, it does not matter when those particular analyte molecules that led to background signal entered the IMR.

Any background measurement value taken at a later time, e.g., at $t=t_4$ or at $t>>t_4$ (a measure of the persistent background), would no longer represent the magnitude of the background at $t=t_2$ and would underestimate the contribution of background signal to the total at the time the background measurement was initiated. This aspect is critical to the determination of so-called tails of measurements, e.g., when an aircraft platform is measuring in an analyte plume and then abruptly exits the plume to analyte-free air. The signal appears to decay as between $t=t_3$ and $t=t_4$ (and beyond) in Fig. 2b. The entirety of this signal is often due to background signal. If this background signal is not subtracted as described herein, the data would be falsely reporting a non-zero concentration (i.e., tail) of the analyte after exiting the plume, which could lead to large errors in measurement-model comparisons that would not be captured by simple uncertainties estimated by

replicate calibrations. Note that when calculating the integral of signal across a plume pass, the same integrated
concentration can be found whether or not the background signal subtraction method is used, provided that a
self-consistent calibration value is applied (see Sect. 3.2).

With the IMR described in this work as well as previous versions, it was found that a 'fast zero'

background measurement of 6 s duration was sufficient to pass through the fast exponential decay (which
typically lasts ~2 s) and capture t=$t_3$ at the start of the slow exponential decay of analyte. The frequency at
which the background measurement needs to be taken depends on the application. For measurement of rapidly
changing analyte concentrations, the background needs to be determined as rapidly as possible to minimize
errors in interpolation of the background. For recent aircraft measurements using this IMR, these 6 s
background measurements were performed once per minute, striking a balance between minimizing
background interpolation errors while maximizing the duty cycle of taking ambient measurements. One could
imagine taking a 6 s background (or shorter, e.g., 4 s) as fast as every 20-30 seconds to capture extremely rapid
changes in some specific circumstances, but information about the same temporal changes in background-
subtracted signal would be lost. Conversely if the analyte concentrations are known to be relatively constant,
e.g., in a laboratory experiment, then the intervals between background determinations could stretch much
longer without leading to substantial interpolation errors. Linear interpolation can be the simplest method,
however other methods could be used depending on specific circumstances. For instance, a relative-
concentration-dependent interpolation may better describe the background signal for a case where a plume
with large concentration gradient was entered and/or exited between background determinations.
**3.2    Wall Interactions and Calibration of Instrument Response**

The previous section discussed accounting for dynamic background signals in the context of

determining accurate gas-phase concentrations in laboratory or field experiments. Also important is to account
for the background signal during instrument response calibrations. When calibrating, a known amount of an
analyte is injected into the instrument, and the amount of raw signal measured per unit analyte is determined.
This raw signal has to be normalized to a constant number of reagent ions, given that the total number of ions
created by an ion source (and thus clusters formed and signal measured) can change with time. Therefore, a
calibration value for this $I^-$ CIMS typically has units of counts per second per $1 \times 10^6$ total reagent ion count
(TRIC) per ppt of analyte, also called normalized counts per second (ncps) per ppt of analyte. When the raw
signal in units of ncps is divided by the calibration value, a concentration in units of ppt is derived. The
calibration value for each analyte must also be determined as a function of the amount of water vapor in the
IMR.

However, in light of the earlier discussion of background signal, considering the signal as units of ncps is

not enough information. The distinction between background-subtracted ncps and background (including
dynamic and persistent background) ncps, which add to total ncps, is necessary. As illustrated in Fig. 2b, when a
constant concentration of $HNO_3$ (approx. 2 ppbv) from a permeation tube was added into the inlet, neither the
background counts per second nor the total counts per second were constant functions of the amount of $HNO_3$
injected. The background-subtracted counts per second were constant, making that value the only properly
deterministic calibration constant that can be applied regardless of the relative amounts of background vs.
background-subtracted signals. Therefore, it is also recommended that the same background subtraction be
performed on both calibration data and field/laboratory measurement data. Also, care should be taken to
ensure that wall steady state is achieved in any tubing that is used to transfer a calibration gas from its source
to the IMR, such as in the PFA Teflon tubing between the $HNO_3$ permeation tube and the IMR used in this work.
This ensures that the flux of $HNO_3$ coming out of that transfer line is the same as the calibrated flux out of the
permeation device.

If background measurements are not performed or are not possible in a certain configuration, an

alternative method may be used in specific circumstances. The calibration constant could be measured as the
total ncps ppt$^{-1}$ during a time equivalent to $t=t_2$, when wall steady state has been achieved. This calibration
constant represents the total (background-subtracted plus background) amount of signal that a given incoming
gas-phase concentration will generate. It applies only at wall steady state, only when environmental conditions
(e.g, RH) are the same as during calibration, and only in a given inlet configuration. Therefore, the calibration
can only be applied to data that has not been background subtracted, and it will only be accurate when wall
steady state has been achieved and environmental conditions are the same as during calibration. For laboratory
measurements, these conditions may be achieved if special care is taken (e.g., flow tubes, oxidation flow
reactors, or chambers operated in reproducible steady-state modes). However, dynamic conditions in field
studies likely preclude this calibration method from being a routinely viable option for analytes with substantial
background signal. The integral of signal across a plume would still be accurate (i.e., mass balance is achieved in
the IMR), but the real concentration would be underestimated at the start of the plume, and overestimated in
the tail of the plume, provided that there are no signal tails from previous plumes still desorbing from surfaces.
If sufficient background measurements were taken during a measurement period, but the calibration constant
applied to that data was calculated using the total signal, the calibration constant can be retroactively
converted to units of background-subtracted ncps ppt$^{-1}$ by finding a suitable time when wall steady state was
achieved during the measurement period. The ratio of background-subtracted signal to total signal during wall
steady state can be derived and multiplied by the total signal calibration constant to obtain the background-
subtracted calibration constant, using the following equation:
$$\frac{Background-subtracted\ ncps}{ppt} = \frac{Total\ ncps}{ppt} \times \frac{(Background-subtracted\ ncps)_{ss}}{(Total\ ncps)_{ss}},\qquad(1)$$

where the subscript ss implies the value at wall steady state. Lastly, it will be important to keep this relationship
between background-subtracted and total calibration constants in mind when comparing experimentally
derived sensitivities to theoretically calculated sensitivities (as in, e.g., Iyer et al., 2016; Lopez-Hilfiker et al.,
2016; Sekimoto et al., 2017). The theoretical calculations will be estimating the total signal per amount of
analyte, without regard to wall effects.

In summary, the accuracy of a calibration constant will depend on how the wall interactions for an

analyte are quantified during calibration and ambient measurements. For sticky compounds with substantial
wall interactions, systematic biases in instrument response, and thus reported concentrations, can easily
approach a factor of 2 (or much more in signal 'tails') without a self-consistent accounting of wall induced
backgrounds during calibrations and measurements.
**4    Quantifying IMR delay times**
**4.1    Chamber measurement methods**

The wall interactions in the IMR designed in this work were characterized through a series of

experiments, including extensive tests performed in the University of Colorado Environmental Chamber Facility
in Boulder, CO. The chamber contained a 20 m$^3$ FEP Teflon bag operated in batch mode. The experimental
method used in this work has been described in more detail in similar experiments designed to characterize wall
interactions in various types of tubing, Teflon bags, and other instrument inlets (Krechmer et al., 2017; Liu et al.,
2019). Briefly, a series of 1-alkanol compounds ($C_6$, $C_8$, $C_9$, $C_{10}$, and $C_{12}$) were injected into the dark chamber

along with methyl nitrite and NO at room temperature. UV blacklights were turned on for 10 s to photolyze methyl nitrite, producing OH radicals through subsequent chemistry (Atkinson et al., 1981). Rapid oxidation of the 1-alkanol compounds until the OH radicals were depleted led to quasi-stable ppt-level concentrations of a range of oxidation products, including hydroxynitrates (HN), dihydroxynitrates (DHN), and dihydroxycarbonyls (DHC) as listed in Table S1. The volatilities of these compounds were estimated using the SIMPOL method (Pankow and Asher, 2008) as in Liu et al., (2019). Chamber air was sampled into the IMR through a 0.75" OD, approximately 8" long PTFE tube. The sample flow through this inlet and into the IMR was 2 slpm, and the ion flow into the IMR was 3 slpm, for a total flow of 5slpm at a constant ~70 Torr in the IMR.

## 4.2 IMR delay times vs. previous designs

The main goal of updating the IMR design as described in Sect. 3.1 was to reduce the measurement artifacts due interactions between analytes and IMR wall surfaces. As described in detail in Sect. 3.2 above, a reduction in wall-induced artifacts leads to improved spatial/temporal accuracy of the measurements, reduced impacts of possible surface chemistry artifacts, and more easily interpretable data. In this section, we describe the measurements used to quantify the improvement achieved in the new design.

In past experiments, wall interactions occurring in lengths of tubing or in IMRs have been quantified using the amount of time required for a signal to decay to 10% of the maximum total signal after wall steady state had been achieved and the signal source was removed (Neuman et al., 1999; Veres et al., 2008; Pagonis et al., 2017; Deming et al., 2019; Liu et al., 2019). When the ambient source of the compound is removed, the background-subtracted signal rapidly decays and all of the remaining signal is due to molecules evaporating or desorbing from the wall surfaces.

To systematically test delay times in the updated IMR design, we employed the recently developed method of sampling a range of HN, DHN, and DHC oxidation products spanning more than five orders of magnitude in volatility. Further details of the experimental setup can be found in related work (Krechmer et al., 2017; Liu et al., 2019). These compounds were allowed to equilibrate with the chamber walls, and sampling from the chamber then provided a constant source of these compounds. Chamber air was sampled through the co-axial IMR into the CIMS until IMR wall steady state was achieved for all compounds. At this point, UHP $N_2$ was injected into the variable orifice upstream of the IMR, removing the source of analyte and starting the measurement of delay times. While the chamber air was dry for all experiments, measurements were

performed with and without adding an estimated 1-2 x $10^{16}$ molec $cm^{-3}$ water vapor directly to the IMR. This
way, the effects of water vapor on the IMR surfaces were probed.

The delay time measurement for one compound, a $C_9H_{19}NO_5$ DHN with an estimated C* of 14.6 µg $m^{-3}$

(which would typically be categorized as a semivolatile organic compound or SVOC), is shown in Fig. 3. Fast zero
measurements (6 s every 1 min) of the background signal were taken prior to the start of the delay time
measurement, illustrating that wall steady state was reached and that approximately 48% of the total signal
was due to the background in the IMR. In other words, half of those analyte molecules that entered the IMR
had interactions with a wall surface prior to desorbing and being sampled at the detector. Once the delay time
measurement started, the signal due to molecules that did not interact with walls rapidly decayed (within
several seconds) followed by the slower decay of the background signal. The amount of time required for the
total signal to drop to within 10% of the persistent background level (which for this compound was essentially
equal to the baseline noise) was measured to be 356 s, or 5.9 min. This DHN is an example of a compound that
would require the fast zero method of background determination in order to achieve temporal/spatial
resolution when sampling variable concentrations such as plumes. Delay times were also determined for the
range of other compounds present in the chamber for both a dry and humidified IMR.

Liu et al. 2019 compiled delay times for the IMRs of several instruments, including a quadrupole proton

transfer reaction MS (q-PTRMS; Pagonis et al., 2017), a Vocus inlet coupled with a time-of-flight MS (Krechmer
et al., 2018), an $I^-$ CIMS using the commercially available IMR (Aerodyne, Inc.) operated at dry conditions by the
Jimenez group, and a custom design similar to the commercially available IMR operated under humidified
conditions by the Ziemann group. The $I^-$ CIMS instruments were tested using the same method and analytes as
in this work, while the delay times for the q-PTRMS and Vocus instruments were measured using a similar
method involving a series of ketones at equilibrium with the walls in a chamber (Pagonis et al., 2017; Deming et
al., 2019). Figure 4 illustrates the delay times measured here in context with the previous results.

In general, the delay times for the co-axial IMR described herein were approximately an order of

magnitude shorter than for the stainless steel IMR under dry conditions, and approximately 5 times shorter
than the similar but humidified stainless steel IMR. The effects of humidity in an IMR appear to depend both on
the material of the IMR as well as the type of analyte. In stainless steel IMRs, increased humidity led to
uniformly shorter delay times for all analytes. However, in our new IMR, humidity led to slightly longer delay

times for DHN and no change for HN. These results illustrate how the interaction between an analyte and a surface can be determined by a complex combination of factors, including the surface type, surface modifications, and functional groups and properties of the analyte.

For both the dry and humidified stainless steel IMRs, results indicated that delay times started trending back towards shorter values at the lowest measured $C^*$ values. This trend is in contrast to the results from the co-axial IMR. Liu et al. (2019) attributed this to irreversible loss of the analyte to the walls, which would decrease the background signal relative to background-subtracted signal. It may be the case that this irreversible loss for species of $C^* < 100$ $\mu$g m$^{-3}$ is unique to those stainless steel IMR surfaces and doesn't occur on the PFA and FEP Teflon surfaces in the co-axial IMR. However, it may also be the case that those lowest volatility compounds had not yet achieved wall steady state with the inlet tubes and stainless steel IMR walls. This would have led to an artificially low amount of background signal relative to background-subtracted signal, causing underestimates of delay times. Successively lower $C^*$ compounds would be further away from steady state for the sampling time prior to the start of the delay measurement, leading to successively more underestimated delay times. If one assumes the linear relationship (in log-log space) observed in the co-axial IMR and for $C^* > 100$ $\mu$g m$^{-3}$ in the stainless steel IMRs would hold for the lower $C^*$ compounds, the delay times in the stainless steel IMRs would reach on order of ~1000 minutes at most, which would become an implausible amount of time to wait for wall steady state to be reached (and for all of the background to decay during the delay measurement) during a batch mode chamber experiment. Also, it would be extremely difficult to ascertain when wall steady state was achieved due to the slow rate of increase of the background signal.

At first glance, extrapolation of results would indicate that the Vocus and q-PTRMS instruments would have one or several orders of magnitude longer respective delay times for the same HN, DHN, and DHC compounds compared with our new IMR. The Vocus and q-PTRMS instruments are designed primarily for $H_3O^+$ ionization chemistry, typically to target a much more volatile set of analyte compounds compared with $I^-$ ionization. They also typically operate with an IMR pressure in the range of 2 Torr, which will greatly enhance the rates of diffusion to the walls compared with the ~70 Torr $I^-$ CIMS IMRs. Both our new IMR and the Vocus have delay times spanning from a second to greater than several minutes over their respective volatility ranges of interest. However, these results indicate that a Vocus-type design would not perform as well for $I^-$ ionization without modifications.

The IMR used in Lee et al. (2018), which employed the same variable orifice with an $H_2O$ vapor addition
port but with turbulent mixing of ions and analyte, was not tested on the CU chamber. However, laboratory
experiments indicate that the delay time for $HNO_3$ under similar humidified conditions in the Lee et al. (2018)
IMR was approximately a factor of three longer than in this new IMR (see Fig. S2), providing a measure of the
improvements between that design and the one presented herein.
**5    Conclusions**
The effects of wall interactions in mass spectrometer inlets and IMRs have been a persistent but
sometimes nebulous concern for as long as researchers have been sampling gases, particularly the lower
volatility and soluble ones often referred to as "sticky" gases. As the importance of such gases to atmospheric
processes like new particle formation/growth and SOA formation continues to be discovered, so does the need
for higher precision and accuracy of the measurements. Recent research has begun to focus on analyte-surface
interactions, including absorption and adsorption processes and how they can affect measurements in IMRs and
in sample tubing. In this work, we introduced a new IMR design with the goal of reducing IMR wall interactions.
This design was informed by the concepts in this and prior research. It sought to minimize wall interactions by
limiting both turbulent and diffusive mixing to the walls, and by choosing wall surfaces that interact least with
the analyte molecules. The new IMR was shown to have delay times that were 3–10 times shorter than previous
IMR versions. This translates to higher signal-to-noise of the background-subtracted signal (i.e., the signal that
did not interact with walls), less influence from possible surface reactions, and easier interpretation of
measured time series.
Since there are a large number of factors affecting wall interactions, many of which are poorly
understood, there has been little ability for researchers across different platforms to apply a uniform treatment
to wall effects. Here, we aimed to provide a common framework of concepts with which the wall interactions in
all instrumental systems could be described and treated. In this framework, the total signal measured at the
detector for a given analyte can be described as originating from the sum of the following two pathways: 1)
some fraction of the molecules do not interact with IMR wall surfaces, and are sampled with a time response
equal to the average residence time in the IMR, and 2) the remaining fraction of molecules interact with the
IMR walls via adsorption/absorption, and are sampled with a delayed time response longer than the average
IMR residence time. We demonstrated a method of using fast zeroing to separate the signal into these parts,

namely the background-subtracted signal and the dynamic plus persistent background signals. The background-subtracted signal is the only part that is a constant function of, and deterministic of, the concentration of analyte entering the IMR as a function of time, and is thus an essential quantity for accurately capturing time-dependence of analyte concentrations. This framework could be adapted to other inlet and instrument configurations. A consistent manner of calibration was also presented.

This IMR design and the characterization of wall interactions represents an improvement over previous low-pressure CIMS techniques used in atmospheric chemistry. Future work could build upon this design, for instance by further decreasing wall interactions. One could also imagine a case where the walls are modified/treated with a method similar to that in Roscioli et al. (2016), but in such a way as to make the walls an irreversible sink for a particular analyte, thereby eliminating the background signal and making the total signal equal to the background-subtracted signal. However, finding a modification technique that would work for the entire range of diverse analyte molecules to which iodide-adduct ionization is sensitive could prove challenging.

To facilitate comparisons and merging of data sets from different instruments, we also encourage the users of all CIMS techniques to adopt the methods for calibration and background subtraction discussed herein when sampling analytes that suffer from wall interactions, and encourage the reporting of all relevant sampling and calibration method details in the publication of such research.

**Data Availability**

All data is available upon request to the authors.

**Author Contributions**

BBP and JAT designed, assembled, and tested the IMR, and determined the framework of wall interactions. BBP, XL and JLJ conducted the characterization experiments at the CU Environmental Chamber facility and participated in the analysis of the data. BBP wrote the manuscript. All authors contributed to revisions of the manuscript.

**Competing Interests**

The authors declare that they have no conflict of interest.

**Acknowledgements**


This research was funded by grant AGS-1652688 from the U.S. National Science Foundation (NSF) and

by grant NA17OAR4310012 from the National Oceanic and Atmospheric Administration (NOAA). The authors
sincerely thank machinist Dennis Canuelle from the University of Washington for his contributions to the design
and manufacture of the IMR. XL and JLJ were supported by the Sloan Foundation Grant 2016-7173, and the US
DOE (BER/ASR) grant DE-SC0016559. We are grateful for many helpful discussions with Demetrios Pagonis,
Jordan E. Krechmer, and other members of the Aerodyne ToF-CIMS User's Group which helped to shape the
material presented herein. We thank Douglas A. Day for experimental support during the chamber
measurements.

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

**Figures**

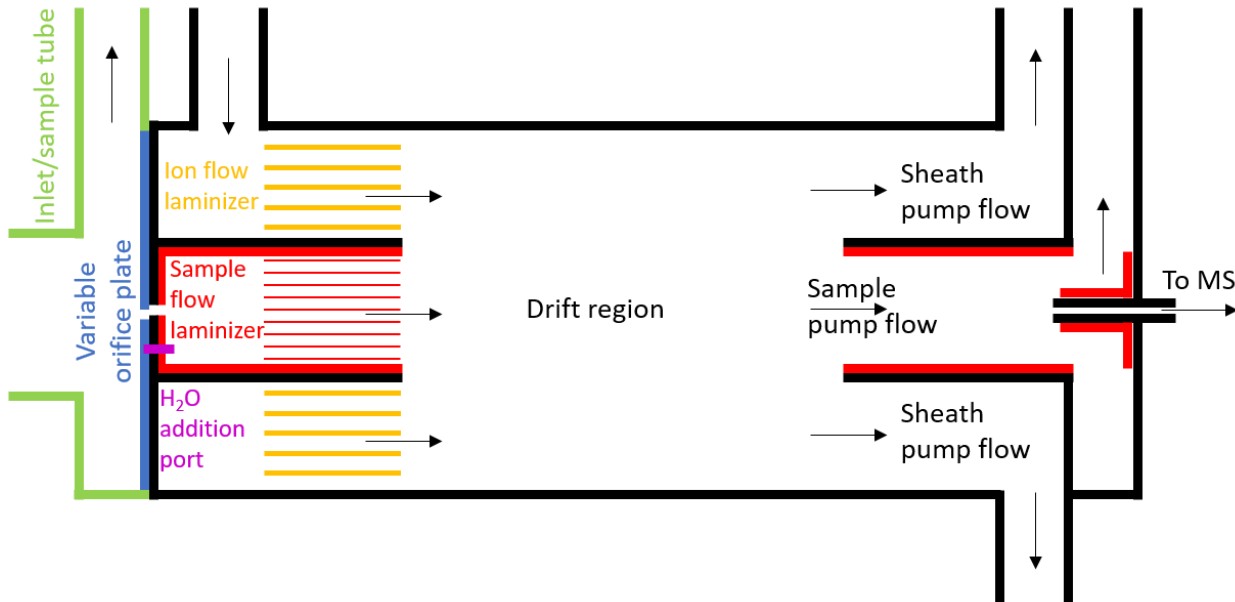


**Figure 1.** Schematic of the new, co-axial, low-pressure IMR design for CIMS. This is a two-dimensional cross
section of the cylindrical IMR along the axis of flow, and it is not to exact scale. Black lines represent stainless
steel surfaces, green and blue lines represent PTFE Teflon, and red/yellow lines represent FEP or PFA Teflon.
Constant mass flow into the IMR is controlled using a variable orifice. Water vapor can be added through a port
in the orifice plate, in order to keep the environmental conditions in the IMR more constant. The sample flow
and ion flow are passed through laminizer elements to limit the effects of turbulent diffusion to the IMR walls.
Ion-molecule adducts are formed via diffusive mixing in the drift region. The ability to enhance the mixing of
ions into the sample flow by applying an electric field between the drift region wall and the exit of the sample
flow laminizer was also included (not shown), but led to only modest enhancement and was not used in the
measurements presented herein. A mirrored pumping scheme also prevents turbulence and limits the effects of
wall interactions. Adducts are sampled through a capillary into the time-of-flight mass spectrometer.


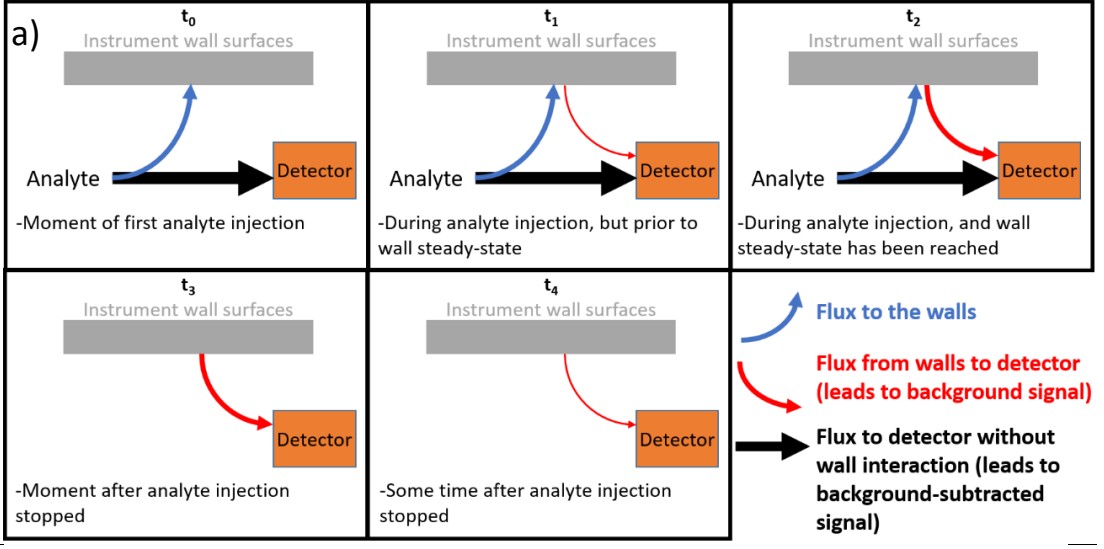


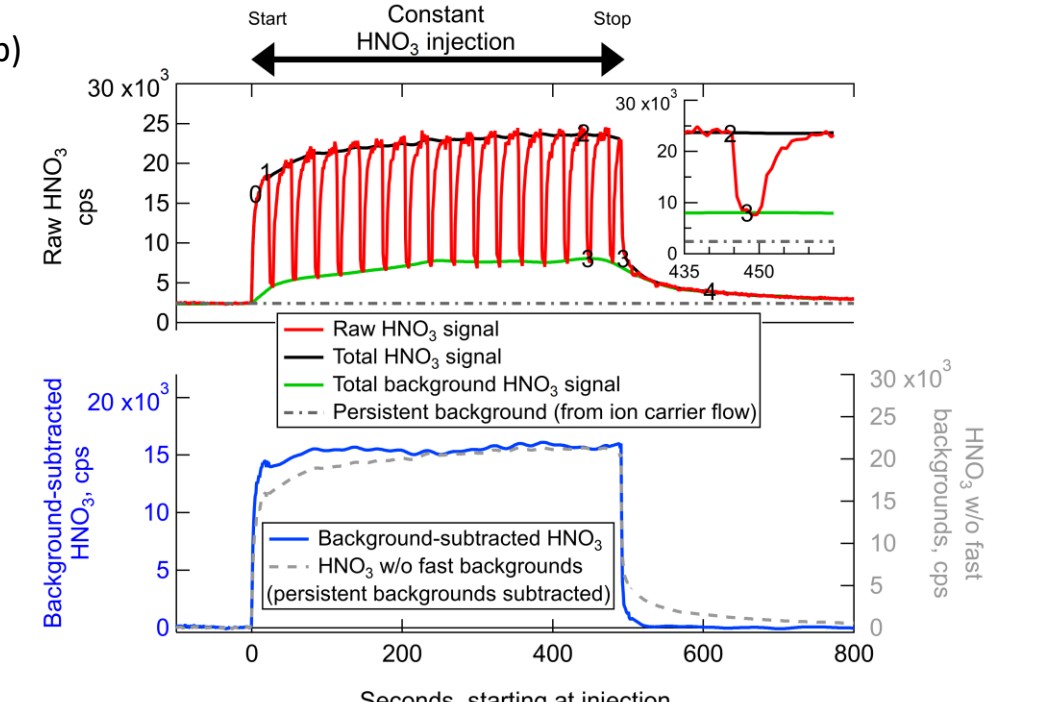

Seconds, starting at injection

**Figure 2.** a) Schematic illustrating how wall interactions affect the measurement of low volatility or polar gases for several experimental conditions, and b) example of the fast zero method of background subtraction for the measurement of constant concentration of ~2 ppbv nitric acid from a permeation tube. The times corresponding to each panel in a) are labeled on the time series in panel b). The bottom of panel b) illustrates the benefits of performing frequent background signal subtractions as opposed to only subtracting the persistent background signal.

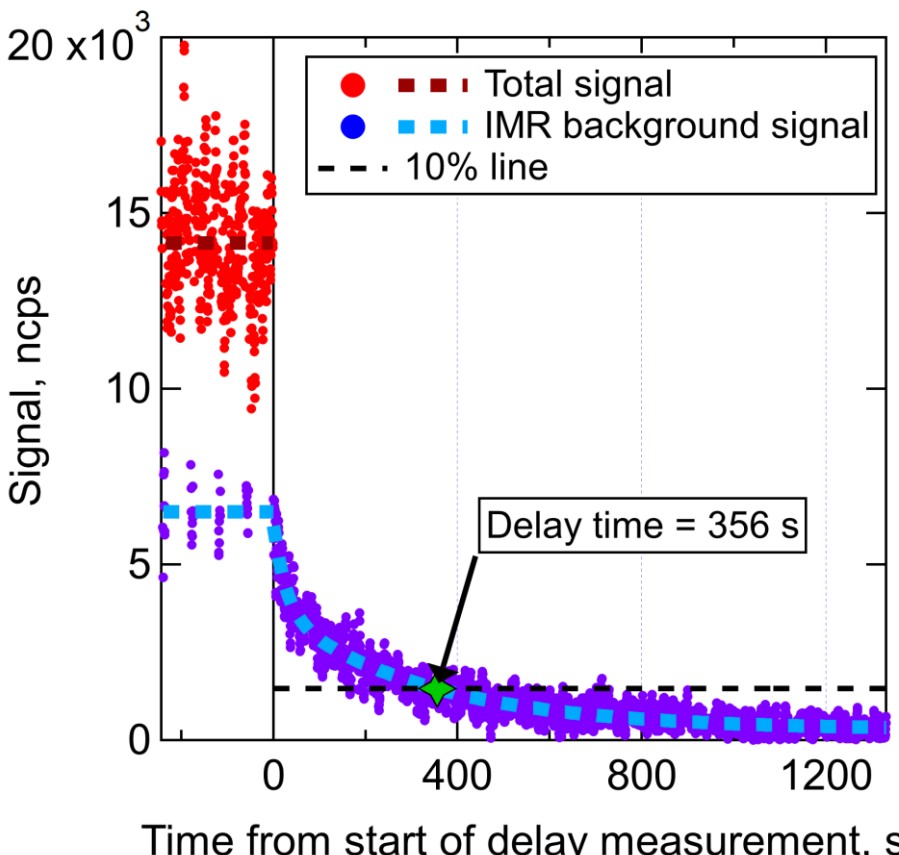

828

**Figure 3.** Delay time measurement of a DHN ($C_9H_{19}NO_5$) in an I⁻ CIMS with the new IMR. Prior to the start of the

delay measurement, wall steady state had been achieved. The total signal is equal to the background-

subtracted signal plus the background signal. Regular background measurements were performed for 6 s of

each 1 min, illustrating that approximately half of the $C_9H_{19}NO_5$ that entered the IMR was interacting with the

walls prior to desorbing and being sampled. The delay time for this DHN, defined as the time required for the

signal to return to 10% of the original value, was determined to be 356 s.

835

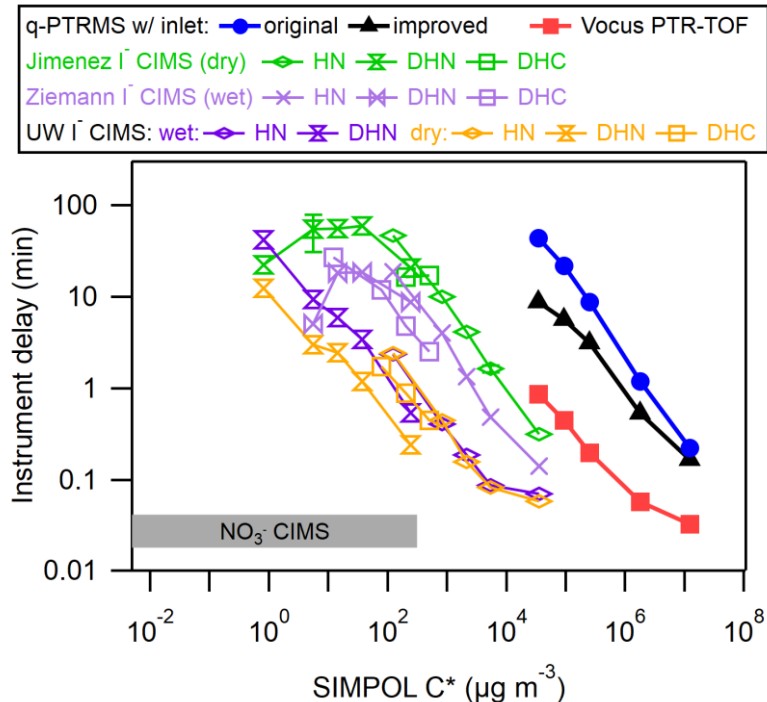

836

**Figure 4.** Delay times for a variety of organic molecules as a function of saturation vapor concentration (C*, µg
m⁻³), compared with previous IMR designs including a q-PTRMS, Vocus PTR-TOF-MS, and several I⁻ CIMS
instruments with different IMRs. The delay time in a nitrate (NO₃⁻) CIMS is also shown for comparison. The
organic molecules are described in Table S1.

841

842