# Peer review of "Performance of a new co-axial ion-molecule reaction region for low-pressure chemical"

_Atmospheric Measurement Techniques, 2019_

## Referee Comment (RC1) · Anonymous Referee #1 · 16 Aug 2019

"Performance of a new co-axial ion-molecule reaction region for low-pressure chemical ionization mass spectrometry with reduced instrument wall interactions" by Palm et al. describes a new CIMS inlet and methods for determining sensitivities and wall losses that can be applied to other CIMS inlet designs. As the authors state, characterizing instrument performance to "sticky" compounds is very important as instruments become more and more capable of their measurement. The article is a pleasure to read, and I have only a few editorial suggestions. Otherwise I recommend publication in AMT.

Minor suggestions:

33: is "relatively" necessary in this sentence (if so, relative to what?)

181 and 224: I recommend "at the inlet to the IMR" rather than "at the start of the IMR" ... in the latter the use of "start" seems not quite appropriate.

184: I recommend modifying this sentence to The flow coming out of both the sample flow laminizer and ion flow laminizer exit in the same plane ..." (that is, the flows are not co-planar).

244: I recommend "applies mainly to analyte..."

249: Is "theoretical" necessary in this sentence?

273: This sentence (to me) is awkward as it suggests surfaces compete for surfaces ... Maybe change to: "Surfaces can also be modified by other analyte molecules, the latter essentially competing for surface sites."

316: The reference to Fig. 3 is not correct, but I also think (since this is actually out of sequence) that the whole parenthetical comment can be eliminated (I don't see how reference to the figure is helpful).

377: correct: "intercation"

441: It may be helpful to note here that there are many studies performed using laminar flow reactors (aka "flow tubes") that, according to this analysis, could be operated in steady state.

478: Is this sentence necessary? I doubt the reader will forget that the inlet was just described.

494: Similarly, the sentence that starts "Further details of the experimental setup..." does not seem necessary.

501: In the sentence: "This way, the surface effects of water vapor on the IMR surfaces were probed." Is the second "surface" needed?

---

## Referee Comment (RC2) · Anonymous Referee #2 · 24 Sep 2019

CIMS has been widely used to measure trace gases. One of the most important artifacts for measuring low volatile gases is the inlet wall effects. The present work designed a low pressure IMR (I- as reagent ion) to reduce the wall impact. Also this study introduces a conceptual framework for considering instrument wall interactions and a measurement protocol to accurately capture the time-dependence of target compound concentrations. The manuscript is well written. I suggest acceptance of this manuscript on AMT.

---

## Author Comment (AC1) · 9 Oct 2019

Response to reviewer comments on "Performance of a new co-axial ion-molecule reaction region for low-pressure chemical ionization mass spectrometry with reduced instrument wall interactions"

Brett B. Palm, Xiaoxi Liu, Jose L. Jimenez, Joel A. Thornton

We thank the reviewers for their comments on our paper. To facilitate the review process, we have copied the reviewer comments in black text. Our responses are in regular blue font. We have responded to all the referee comments and made alterations to our paper (**shown in bold text**).

Anonymous Referee #1

"Performance of a new co-axial ion-molecule reaction region for low-pressure chemical ionization mass spectrometry with reduced instrument wall interactions" by Palm et al. describes a new CIMS inlet and methods for determining sensitivities and wall losses that can be applied to other CIMS inlet designs. As the authors state, characterizing instrument performance to "sticky" compounds is very important as instruments become more and more capable of their measurement. The article is a pleasure to read, and I have only a few editorial suggestions. Otherwise I recommend publication in AMT.

Minor suggestions:

33: is "relatively" necessary in this sentence (if so, relative to what?)

We have removed the word "relatively" from the sentence, as it is not needed.

181 and 224: I recommend "at the inlet to the IMR" rather than "at the start of the IMR" . . . in the latter the use of "start" seems not quite appropriate.

As we are defining the word "inlet" as referring to the sampling tube, we would prefer not to use the same word when referencing the IMR. Instead, we have changed the text at line 181 from "at the start of the IMR" to **"into the IMR"**, and at line 224 from "to the start of the IMR" to **"to the IMR"**.

184: I recommend modifying this sentence to "The flow coming out of both the sample flow laminizer and ion flow laminizer exit in the same plane . . ." (that is, the flows are not co-planar).

We have modified the text as suggested.

244: I recommend "applies mainly to analyte. . ."

We have modified the text as suggested.

249: Is "theoretical" necessary in this sentence?

We thank the reviewer for their question, but we believe the word "theoretical" helps to make clear to the reader that we are introducing a framework of concepts that can be applied to other experimental data, rather than just specifically to the data presented in Fig. 2.

273: This sentence (to me) is awkward as it suggests surfaces compete for surfaces . . . Maybe change to: "Surfaces can also be modified by other analyte molecules, the latter essentially competing for surface sites."

We have modified the text to read **"Surfaces can also be modified by other analyte molecules, which essentially act in competition for surface sites."**

316: The reference to Fig. 3 is not correct, but I also think (since this is actually out of sequence) that the whole parenthetical comment can be eliminated (I don't see how reference to the figure is helpful).

We have deleted the text "(shaded box in Fig. 3)". To further ensure that the figures are referenced in the correct order, we have changed the text at line 338 from "As illustrated in Fig. 4 and discussed more in Sect. 3.3,…" to **"As discussed further in Sect. 3.3,…"**. We have also corrected an error by changing the reference in line 385 from "Fig. 4b" to **"Fig. 2b"**.

377: correct: "intercation"

We have changed the text from "intercation" to **"interaction"**.

441: It may be helpful to note here that there are many studies performed using laminar flow reactors (aka "flow tubes") that, according to this analysis, could be operated in steady state.

We have modified the text to read "For laboratory measurements, these conditions may be achieved if special care is taken **(e.g., flow tubes, oxidation flow reactors, or chambers operated in reproducible steady-state modes).**"

478: Is this sentence necessary? I doubt the reader will forget that the inlet was just described.

We have deleted the sentence as suggested.

494: Similarly, the sentence that starts "Further details of the experimental setup. . ." does not seem necessary.

We have modified the text from "Further details of the experimental setup can be found in Sect. 4.1 and in related work (Krechmer et al., 2017; Liu et al., 2019)." to **"Further details of the experimental setup can be found in related work (Krechmer et al., 2017; Liu et al., 2019)."**

501: In the sentence: "This way, the surface effects of water vapor on the IMR surfaces were probed." Is the second "surface" needed?

We have removed the first instance of "surface" from this sentence.

---

## Author Comment (AC2) · 9 Oct 2019

Response to reviewer comments on "Performance of a new co-axial ion-molecule reaction region for low-pressure chemical ionization mass spectrometry with reduced instrument wall interactions"

Brett B. Palm, Xiaoxi Liu, Jose L. Jimenez, Joel A. Thornton

We thank the reviewers for their comments on our paper. To facilitate the review process, we have copied the reviewer comments in black text. Our responses are in regular blue font. We have responded to all the referee comments and made alterations to our paper (**shown in bold text**).

Anonymous Referee #2

CIMS has been widely used to measure trace gases. One of the most important artifacts for measuring low volatile gases is the inlet wall effects. The present work designed a low pressure IMR (I- as reagent ion) to reduce the wall impact. Also this study introduces a conceptual framework for considering instrument wall interactions and a measurement protocol to accurately capture the time-dependence of target compound concentrations. The manuscript is well written. I suggest acceptance of this manuscript on AMT.

We thank the reviewer for their comments.